# Development of a Vaccine against SARS-CoV-2 Based on the Receptor-Binding Domain Displayed on Virus-Like Particles

**DOI:** 10.3390/vaccines9040395

**Published:** 2021-04-16

**Authors:** Lisha Zha, Xinyue Chang, Hongxin Zhao, Mona O. Mohsen, Liang Hong, Yuhang Zhou, Hongquan Chen, Xuelan Liu, Jie Zhang, Dong Li, Ke Wu, Byron Martina, Junfeng Wang, Monique Vogel, Martin F. Bachmann

**Affiliations:** 1International Immunology Centre, Anhui Agricultural University, Hefei 230036, China; zhalisha@ahau.edu.cn (L.Z.); hongliang@ahau.edu.cn (L.H.); chqchq@ahau.edu.cn (H.C.); xuelan.liu@dbmr.unibe.ch (X.L.); 2Department of Rheumatology and Immunology, University Hospital Bern, 3010 Bern, Switzerland; xinyue.chang@dbmr.unibe.ch (X.C.); mona.mohsen@dbmr.unibe.ch (M.O.M.); Monique.vogel@dbmr.unibe.ch (M.V.); 3Department of BioMedical Research, University of Bern, 3012 Bern, Switzerland; 4High Magnetic Field Laboratory, Chinese Academy of Sciences, Hefei 230031, China; zhx@hmfl.ac.cn; 5Saiba AG, 8808 Pfäffikon, Switzerland; 6Shandong H&Z Lifescience Gmbh, Yantai 264000, China; huangtsian@163.com (Y.Z.); jie_zhang@sinobiological.com (J.Z.); 7Beijing Key Laboratory of Monoclonal Antibody Research and Development, Beijing 100176, China; dong_li@sinobiological.com; 8Institute of Risk Analysis, Prediction and Management, Academy of Interdisciplinary and Advanced Studies, Southern University of Science and Technology, Shenzhen 518055, China; wuk@sustech.edu.cn; 9Artemis Reserch One Heath Foundation, 7100 Delft, The Netherlands; b.martina@artemis-biosupport.com; 10Jenner Institute, Old Road Campus, University of Oxford, Roosevelt Drive, Oxford OX3 7BN, UK

**Keywords:** COVID-19, vaccine, virus-like particle, CuMV_TT_–RBD, BLI

## Abstract

The ongoing coronavirus disease (COVID-19) pandemic is caused by a new coronavirus (severe acute respiratory syndrome coronavirus type 2 (SARS-CoV-2)) first reported in Wuhan City, China. From there, it has been rapidly spreading to many cities inside and outside China. Nowadays, more than 110 million cases with deaths surpassing 2 million have been recorded worldwide, thus representing a major health and economic issues. Rapid development of a protective vaccine against COVID-19 is therefore of paramount importance. Here, we demonstrated that the recombinantly expressed receptor-binding domain (RBD) of the spike protein can be coupled to immunologically optimized virus-like particles derived from cucumber mosaic virus (CuMV_TT_). The RBD displayed CuMV_TT_ bound to ACE2, the viral receptor, demonstrating proper folding of RBD. Furthermore, a highly repetitive display of the RBD on CuMV_TT_ resulted in a vaccine candidate that induced high levels of specific antibodies in mice, which were able to block binding of the spike protein to ACE2 and potently neutralize SARS-CoV-2 virus in vitro.

## 1. Introduction

Coronavirus disease (COVID-19) is a disease caused by a novel coronavirus closely related to viruses causing Severe Acute Respiratory Syndrome (SARS) and Middle East Respiratory Syndrome (MERS). Similar to the disease caused by the other two viruses, COVID-19 mainly manifests symptoms in the lung and causes cough and fever [1]. The disease COVID-19 is less severe than SARS and MERS, which is advantageous per se but leads to more rapid spread because infected asymptomatic individuals (mostly pre-symptomatic) may transmit the virus [2]. To prevent the further spread of COVID-19, primary efforts focus on confinement, with physical distancing and multiple further measures preventing infection [3,4]. Nowadays, many investigations aim at defining optimal strategies to limit viral transmission while simultaneously permitting business and social life activities [5]. Among them, one strategy is to induce a rapid onset of immune protection by using vaccines against COVID-19. Different vaccine candidates have already been approved for general or emergency use or are in clinical trials all targeting the spike protein of severe acute respiratory syndrome coronavirus type 2 (SARS-CoV-2) [6,7]. Most vaccines are based on either viral vectors (e.g., adenovirus, University of Oxford/AstraZeneca, Gamaleya Sputnik V, Johnson & Johnson) or mRNA (Moderna, CureVac, Pfizer-BioNTech), as such vaccine candidates can be rapidly produced under Good Manufacturing Practice (GMP) conditions and no particular design is required. Both types of vaccines encode the spike protein (or part of it). RNA vaccines, formulated in lipid nanoparticle [8], and recombinant adenovirus-based vaccines [9] are currently in use for vaccination in several countries. Other vaccines are based on whole virus (live-attenuated or inactivated), and recently, two vaccines consisting of inactivated SARS-CoV-2 plus alum adjuvant (Sinovac Biotech and Sinopharm) have been approved for general use in China [10,11]. However, despite the progress of these vaccines, there are many obstacles hindering their rapid and efficient use, especially in developing countries, such as storage requirements at low temperatures of −80 °C (Pfizer/BioNTech vaccine) [12] and the high price of some of them [13]. Hence, next-generation vaccines should be more affordable, and handling should be easier compared with the current RNA-based vaccines. In addition, as SARS-CoV-2 may stay with us for many years or decades, it would be beneficial if the vaccine could be applied multiple times to the same individual, which is a major problem for adenoviruses. An additional important requirement for effective vaccines is their efficacy in old people and all ethnicities, despite comorbidities [14]. Because of these variables, not all vaccines may be ideal in all situations, and there is room for different vaccine types optimized for specific settings.

Virus-like particles (VLPs) may offer an additional attractive possibility for COVID-19 vaccine development, as they have already been used as successful vaccine delivery platforms [15,16]. They can be engineered to display epitopes of foreign viruses on their surface with an optimized spacing of about 5 to 10 nm, rendering those epitopes highly immunogenic. We have previously shown that antigens displayed on virus-like particles (VLPs) induce high levels of antibodies in all species tested, including humans [17]. More recently, we have developed an immunologically optimized VLP platform based on cucumber mosaic virus. These cucumber mosaic virus (CuMV_TT_) VLPs incorporate a universal T cell epitope derived from tetanus toxin, providing pre-existing T cell help. In addition, these VLPs package bacterial RNA, which is a ligand for toll-like receptor 7/8 and serves as potent adjuvant by engaging these innate receptors in specific B cells [18]. Using antigens displayed on these VLPs, it is possible to induce high levels of specific antibodies in mice, rats, cats, dogs, and horses and treat diseases such as atopic dermatitis in dogs or insect bite hypersensitivity in horses [18,19,20]. In comparison with other vector-based vaccines, such as adeno- or adeno-associated viruses, VLPs do not need to infect cells for antigen expression. Hence, in contrast to the above-mentioned vectors, VLPs do not induce neutralizing antibodies against themselves and therefore can be applied multiple times. VLPs have several advantages, including a structure similar to viruses that can induce a strong immune response, the versatility of VLPs in antigen presentation, and most importantly, a good safety profile as no infection is required and no pre-existing immunity may result in undue immunological responses [21].

Here, we used CuMV_TT_-VLP as a vaccine platform to generate an immune response against the SARS-CoV-2 spike protein. The spike protein of SARS-CoV-2 shows a sequence similarity of 76% to 78% with the spike protein of (severe acute respiratory syndrome coronavirus type 1 (SARS-CoV-1) [22], and both viruses share the same receptor, which is angiotensin-converting enzyme 2 (ACE2) [23,24]. The receptor-binding domain (RBD) of the SARS spike protein binds to ACE2 and is an important target for neutralizing antibodies [25,26]. By analogy, the RBD of the SARS-CoV-2 spike protein is also the target of neutralizing antibodies, blocking the interaction of the virus with its receptor [27]. Therefore, we generated a CuMV_TT_–RBD vaccine by chemically coupling the RBD of SARS-CoV-2 on the CuMV_TT_-VLP. We show here that the vaccine is highly immunogenic in mice and is able to elicit antigen-specific antibodies with a neutralizing effect on SARS-CoV-2.

## 2. Material and Methods

### 2.1. Protein Expression and Purification

The SARS-CoV-2 receptor-binding domain (RBD) was expressed using Expi293F cells (Gibco, Thermo Fisher Scientific, Waltham, MA, USA). The RBD (residues Arg319-Phe541) with an N-terminal IL-2 signal peptide for secretion and a 6-His-tag for purification was inserted into a pTwist CMV BetaGlobin WPRE Neo vector (Twist Bioscience, San Francisco, CA, USA). The construct (RBD–His Tag) was transformed into bacterial XL-1 Blue competent cells, and 50 μg plasmid was then transfected into Expi293F cells at a density of 3 × 10^6^ cells/mL in a 250 mL shaking flask using the ExpiFectamine 293 Transfection Kit (Gibco, Thermo Fisher Scientific, Waltham, MA, USA). The supernatant of cell culture containing the secreted RBD was harvested 96 h after infection, dialyzed with PBS. RBD was captured by HisTrap HP column (GE Healthcare, Wauwatosa, WI, USA). Fractions containing the RBD were collected, concentrated, and buffer-exchanged to PBS using Vivaspin 20 5KDMWCO spin column (Sartorius Stedim Switzerland AG, Tagelswangen, Switzerland). Human ACE2 protein His Tag was purchased from Sino Biological, Beijing, China. Human ACE2 fused to mouse IgG2a Fc protein was given by PD Dr. Alexander Eggel (University Clinic of Rheumatology and Immunology, Inselspital, Bern, Switzerland), who obtained the plasmid from Prof. Peter Kim (Stanford University, Stanford, CA, USA).

### 2.2. Production of CuMV_TT_-VLP

The production of CuMV_TT_-VLP was described in detail in Zeltins et al. [28]. Briefly, *Escherichia coli* C2566 cells (New England Biolabs, Ipswich, MA, USA) were transformed with the CuMV_TT_ coat protein (CP) gene-containing plasmid pET CuMV_TT_. The expression was induced with 0.2 mM isopropyl-β-D-thiogalactopyranoside (IPTG). The resulting biomass was collected by low-speed centrifugation and was frozen at −20 °C. After thawing on ice, the cells were suspended in the buffer containing 50 mM sodium citrate, 5 mM sodium borate, 5 mM EDTA, and 5 mM 2-mercaptoethanol (pH 9.0, buffer A) and were disrupted by ultrasonic treatment. Insoluble proteins and cell debris were removed by centrifugation (13,000 rpm, 30 min at 5 °C, JA-30.50 Ti rotor, Beckman, Palo Alto, CA, USA). The soluble CuMV_TT_ CP protein in clarified lysate was pelleted using saturated ammonium sulfate (1:1, vol/vol) overnight at 4 °C. Soluble CuMV_TT_ CP-containing protein solution was separated from the cellular proteins by ultracentrifugation in a sucrose gradient (20–60% sucrose; ultracentrifugation at 25,000 rpm for 6 h at 5 °C (SW28 rotor, Beckman)). After dialysis of CuMV_TT_-containing gradient fractions, VLPs were concentrated using ultracentrifuge (TLA100.3 rotor, Beckman, at 72,000 rpm for 1 h, +5 °C) or by ultrafiltration using Amicon Ultra 15 (100 kDa; Merck Millipore, Cork, Ireland).

### 2.3. Generation of the Vaccine CuMV_TT_–RBD

The RBD was conjugated to CuMV_TT_ using the cross-linker succinimidyl 6-(beta-maleimidopropionamido) hexanoate (SMPH) (Thermo Fisher Scientific, Waltham, MA, USA) at 7.5 molar excess to CuMV_TT_ for 30 min at 25 °C. The coupling reactions were performed with molar ratio RBD/CuMV_TT_ (1:1) by shaking at 25 °C for 3 h at 1200 rpm on a DSG Titertek (Flow Laboratories, Irvine, UK). Unreacted SMPH and RBD proteins were removed using Amicon Ultra 0.5, 100 K (Merck Millipore, Burlington, MA, USA). VLP samples were centrifuged for 2 min at 14,000 rpm for measurement on ND-1000. Coupling efficiency was calculated by densitometry (as previously described for the IL17A-CuMV_TT_ vaccine [28]), with a result of approximately 20% to 30% efficiency. 

### 2.4. Electron Microscopy

The integrity of the CuMV_TT_–RBD was assessed by transmission electron microscopy (TEM) as follows: 5 μL of vaccine suspension was adsorbed on glow-discharged and carbon-coated 400 mesh copper grids (Plano, Wetzlar, Germany) for 1 min. After washing them three times by dipping in pure water, the grids were stained with 2% uranyl acetate solution (Electron Microscopy Science, Hatfield, PA, USA) for 45 s. The excess fluid was removed by gently pushing them sideways to filter paper. Samples were then examined with a transmission electron microscope (Tecnai Spirit, FEI, Hillsboro, OR, USA) at 80 kV and equipped with a digital camera (Veleta, Olympus, Münster, Germany).

### 2.5. Mice 

*BALB/c* mice at the age of 7 weeks were purchased from Envigo (Horst, The Netherlands) and kept at the Specific Pathogen Free (SPF) animal facility of the University of Bern (Department of Biomedical Research). All animals were treated for experimentation according to protocols approved by the Swiss Federal Veterinary Office.

### 2.6. Vaccination

Female naive BALB/c mice (8–12 weeks old, five mice per group) were immunized by subcutaneous injection with an optimal dose of 40 µg of either CuMV_TT_–RBD, mixture of 30 µg CuMV_TT_ and 10 µg of RBD (total amount of 40 µg), or Tris buffer (20 mM Tris, 5 mM EDTA, pH 8.0) as control. The boost of the vaccination was conducted at 24 days after prime vaccination. Serum was collected for ELISA analysis on days 14, 21, 31, and 38 after prime vaccination.

### 2.7. Direct ELISA

To assess the immunogenicity of the CuMV_TT_–RBD vaccine, Corning half-area 96-well plates (Sigma-Aldrich, Buchs, Switzerland) were coated with 1 μg/mL RBD–His or spike protein (Sino Biological, Beijing, China). The plates were then blocked with PBS–0.15% casein at room temperature for 2 h, followed by incubation with the sera of all immunized mice, which were threefold serial-diluted starting with a 1:20 dilution and incubated for 1 h. Afterwards, horseradish peroxidase (HRP)-labelled goat anti-mouse IgG-POX antibody (Jackson ImmunoResearch, West Grove, PA, USA) was incubated for 1 h at room temperature. Finally, the developing solution containing 3,3′,5,5′-tetramethylbenzidine (TMB) was added and stopped by 1 M H_2_SO_4_ solution. The amount of specific antibody was measured by OD_450nm_.

### 2.8. Sandwich ELISA

To test whether the RBD coupled on CuMV_TT_ can be recognized by anti-RBD antibodies and ACE2, plates were first coated with 5 μg/mL anti-CuMV_TT_ antibody (house-made monoclonal antibody from hybridoma) to capture the CuMV_TT_ vaccine, followed by incubation with different concentrations of the CuMV_TT_–RBD vaccine. Then human anti-RBD antibody (Sanyou Biopharmaceuticals, Shanghai, China) or biotinylated ACE2 (Sino Biological, Beijing, China) was added and incubated on plates for 1 h at room temperature. Next, HRP-labelled goat anti-human IgG (Nordic-MUbio, Susteren, The Netherlands) or HRP-labelled streptavidin (Thermo Fisher Scientific) was added for 1 h at room temperature. The plates were developed as described above using a TMB substrate. The readout of OD_450nm_ was plotted in accordance with increasing CuMV_TT_–RBD concentrations.

### 2.9. Serum Competitive ELISA

The antibody competitive binding activities of the serum were assayed by ELISA. ACE2 (1 μg/mL) was incubated in a 96-well plate overnight at 4 °C. After incubation, the plate was blocked with 2% BSA for 2 h at 37 °C and then was washed five times with PBS containing 0.05% Tween 20. BSA was used as negative control, followed by the addition of a mixture of 40-fold diluted serum and RBD–His (0.15 μg/mL), followed by incubation for 30 min with gentle shaking at 37 °C. Plates were washed five times with PBS containing 0.05% Tween 20 (PBT), followed by 100 µL of HRP-labelled anti-mFc antibody (diluted 1:5000 in PBT buffer), incubated 30 min with gentle shaking. Plates were washed five times with PBT buffer and developed with 100 µL of freshly prepared TMB substrate. Reaction was stopped with 100 µL of 1.0 M H_3_PO_4_ and read spectrophotometrically at 450 nm in a microtiter plate reader.

### 2.10. BLI-Based Competitive Assay

The ability of the sera of the immunized mice to compete with ACE2 for binding to the RBD was tested in a sandwich format assay using biolayer interferometry (BLI) on the Octet RED96e (ForteBio, Fremont, CA, USA). Anti-penta-His (HIS1K) biosensors were loaded for 10 min with the RBD at a concentration of 7.5 μg/mL in BLI assay buffer (PBS, 0.1% BSA, 0.02% Tween 20). The sensor tips were then dipped in wells containing samples (diluted 1:20 in BLI buffer) from mice vaccinated with CuMV_TT_–RBD and for control purposes from mice vaccinated with either a mixture of equal amounts of CuMV_TT_ and RBD or buffer alone (20 mM Tris/5 mM EDTA pH 8.0). Association was followed for 10 min. To assess whether the sera can inhibit the binding of ACE2 to RBD, tips were then placed in wells containing ACE2 at a concentration of 50 nM, and association was measured for 10 min. For control, two additional tips with BLI buffer were used, one for baseline and one without serum sample to determine the binding of ACE2 alone. The response data were normalized using ForteBio data analysis software version 1.2.0.1.55.

### 2.11. Generation of Pseudovirus

Pseudovirus expressing the SARS-CoV-2 spike protein was produced by a lentivirus second-generation packaging system [28]. Plasmids of pwpxl-luc, HIV-1 PSD, and pCMV3 containing the SARS-CoV-2 spike gene were cotransfected into 7 × 10^5^ 293LT cells using Sinofection (Sino Biological, Beijing, China). The medium was replaced with fresh DMEM containing 10% FBS after overnight incubation. Supernatants containing pseudovirus were collected 48 and 72 h after transfection and filtered using a 0.45 μm filter syringe. All filtered supernatants were collected together and stored at −80 °C until use. 

### 2.12. Titration of Pseudovirus

The 293T-ACE2 cells, which stably express ACE2 receptors on the cell membrane, were prepared by transfection of the ACE2 gene into 293T cells using lentivirus system. The 293T-ACE2 cells were generated by transfecting 293FT cells with 500 ng MLV GagPol expression vector, 400 ng of retroviral transfer vector pQCXIP-ACE2, and 100 ng of VSV-G expression vector. Viral medium was used to transduce 293FT cells or the 293FT sensor cell line, and cells were selected with puromycin (1 μg/mL) beginning 2 days postransduction and were maintained until control cells were all eliminated by puromycin system. Pseudoviruses and a series of 10-fold diluted pseudoviruses (diluted with DMEM) prepared above were added to the 293T-ACE2 cells (3 × 10^4^ cells/well) with 100 μL polybrene (16 μg/mL). Two replicates for each pseudovirus concentration were set as control. After 48 h, the infection was monitored using the Luciferase Assay System (Promega, Madison, WI, USA). Titer was calculated based on serial dilutions of pseudovirus. The dilution of approximately 10% pseudovirus was selected as titer for neutralization assay.

### 2.13. Pseudovirus Neutralization Assay

The mouse serum samples (2 μL) were diluted 1:10, and then mixed with an equal volume of pseudovirus stock (the suitable concentration of pseudovirus was confirmed in the titration assay as described in Section 2.12). After incubation at 37 °C for 1 h, the mixture was inoculated on the 293T-ACE2 cells (3 × 10^4^ cells/well). At the same time, pseudovirus + DMEM medium was set as a positive control, and DMEM medium only was set as a negative control, each sample containing three replicates. The cells were incubated at 37 °C, 5% CO_2_ afterwards. After 72 h, serum neutralization was measured by luciferase activity of infected pseudovirus. An amount of 100 μL of supernatant of each well was collected, and 100 μL of luciferase substrate was added to each of the sample. Two minutes after incubation at room temperature, 150 μL of lysate was transferred to 96-well plates for the detection of luminescence using a microplate luminometer. The positive well was determined as 10-fold relative luminescence unit (RLU) values higher than the cell background. The percentage of vaccinated serum neutralization was determined by normalizing the positive control value as 100%. Briefly, pseudovirus-containing supernatants were respectively incubated with serially diluted mouse sera at 37 °C for 1 h before adding to target 293T-ACE2 cells pre-plated in 96-well culture plates (3 × 10^4^ cells/well). Twenty-four hours later, the cells were refreshed in fresh medium, which was followed by lysing cells 72 h later using cell lysis buffer and transferring the lysates into 96-well luminometer plates. Luciferase substrate was added to the plates, and relative luciferase activity was determined. The corresponding neutralizing antibody concentration was calculated.

### 2.14. SARS-CoV-2 Neutralization Assay

Serum samples were first heat-inactivated at 56 °C for 30 min. Subsequently, the samples were diluted twofold starting from 1:20 dilution until 1:160. A suspension of 100 TCID50 of SARS-CoV-2/ABS/NL20 was added to each well and incubated for 1 h at 37 °C. Afterwards, the mixtures were added on a monolayer of Vero-E6 cells and incubated for additional 4 days at 37 °C. Four days later, the plates were stained with crystal violet, and the wells were inspected for the presence of cytopathic effect (CPE). Titer was expressed as the highest dilution of the serum that fully inhibits the formation of CPE.

### 2.15. Data and Statistical Analysis

All statistical tests were performed using GraphPad Prism 6.0 (GraphPad Software, Inc., San Diego, CA, USA). ELISA data in graphs and BLI data are shown as area under the curve for each individual mouse. Significant analysis was done by unpaired two-tailed Student‘s *t*-test and displayed as *p* ≤ 0.05 (*), *p* ≤ 0.01 (**), *p* ≤ 0.005 (***), *p* ≤ 0.001 (****).

## 3. Results

### 3.1. Generation of the CuMV_TT_-RBD Vaccine

To generate a COVID-19 vaccine candidate, we displayed the RBD domain of the SARS-CoV-2 spike protein on the repetitive surface of CuMV_TT_ (Figure 1A). To this end, the RBD–His protein was chemically coupled to the surface of CuMV_TT_ using the well-established chemical cross-linker SMPH [18]. The coupling efficiency of RBD was analyzed by SDS-PAGE as shown in Figure 1B. Densitometric analysis confirmed the efficient coupling of the RBD to the CuMV_TT_, resulting in a coupling efficiency of about 20% to 30% (i.e., ca 40–50 RBDs per VLP). Therefore, approximately 8 to 12 μg of RBD was present per 40 μg of the vaccine. We next examined the natural conformation of the RBD on the surface of CuMV_TT_ by means of recognition by monoclonal anti-RBD antibody and recombinant ACE2 (Figure 1C). We observed that the RBD is well recognized by the anti-RBD antibody and also strongly binds to ACE2, supporting the notion that the RBD has the right conformation able to bind ACE2. Finally, we used VLP electron microscopy for analysis of the morphology of the CuMV_TT_–RBD vaccine, demonstrating that VLPs have the right overall structure and integrity (Figure 1D, yellow stars). In summary, the CuMV_TT_–RBD vaccine candidate was successfully generated, and the RBD was displayed in proper conformation.

### 3.2. Immunogenicity of the RBD–CuMV_TT_ Vaccine

To test the immunogenicity of the vaccine candidate, mice were immunized with CuMV_TT_–RBD, a mixture of equal amounts (total 40 μg) of 30 μg CuMV_TT_ and 10 μg RBD or buffer as control (Figure 2A). The binding specificities of the serum samples were determined either by ELISA using the RBD–His and spike protein or by biolayer interferometry (BLI) using only the RBD–His protein as coating antigens (31 days after first immunization). Our data demonstrated that coupling the RBD to VLPs dramatically increased the immunogenicity of the RBD. As shown in Figure 2B,C, antibody titers are substantially increased in mice immunized with CuMV_TT_–RBD, especially after booster immunization. Vaccination of mice with a mixture of CuMV_TT_ and RBD or with buffer only induced low or no antibody response, respectively, underlining the importance of repetitive display on VLPs.

### 3.3. Serum Antibodies Compete with ACE2-Fc for RBD Binding

Next, we performed competitive assays to determine whether the anti-RBD sera from immunized mice can compete with ACE2 for binding to the RBD. We developed first a competitive ELISA where ACE2 was immobilized on plates, followed by the addition of the sera of immunized mice in the presence of the RBD. As shown in Figure 3A, the sera of mice immunized with CuMV_TT_–RBD obtained 31 days after first vaccination were able to strongly inhibit RBD binding to ACE2. For comparison, sera from mice immunized with either a mixture of CuMV_TT_ and RBD or buffer (Tris buffer) did not show any significant competition for the binding to the RBD. In parallel, we performed a competitive BLI assay using a different setup. In this assay, we assessed the capacity of the sera of immunized mice to compete with ACE2 for binding to the RBD. Consistent with the ELISA results, only the sera of mice immunized with CuMV_TT_–RBD were able to block binding of ACE2 to RBD (Figure 3B). These results indicate that immunization with CuMV_TT_–RBD induces anti-RBD antibodies, recognizing an epitope that overlaps with or in the vicinity of the ACE2 binding site of the SARS-CoV-2 RBD and, therefore, is able to block the interaction of the RBD with the receptor ACE2, the basis of viral neutralization.

### 3.4. Neutralizing Activity of the Anti-RBD Antibodies

For most viruses, the best correlate of protection upon vaccination is viral neutralization [29]. As shown above, the antibodies induced by vaccination with CuMV_TT_–RBD were able to block viral interaction with ACE2. To assess the actual viral neutralization, we set out to determine the neutralization of pseudotyped retroviruses as well as actual SARS-CoV-2. To this end, we generated pseudotyped retroviruses expressing the SARS-CoV-2 spike protein and luciferase for the quantification of infection (Figure 4A). Using these viruses, the neutralizing capacity of the sera from immunized mice was assessed on ACE2-transfected cells (Figure 4B), directly demonstrating high antiviral neutralizing activity of the sera of mice immunized with CuMV_TT_–RBD, whereas no neutralization was observed when viruses were incubated with sera from mice immunized with the RBD mixed with CuMV_TT_ (Figure 4C). Furthermore, the capacity of CuMV_TT_–RBD-vaccinated mice sera to neutralize actual SARS-CoV-2 was tested next on Vero cells using the cytopathic effect of the virus as a readout. CuMV_TT_–RBD immune sera showed high neutralizing titers with the ability to fully block viral replication for 4 days. These titers were much higher than those of mice immunized with a mixture of CuMV_TT_ and RBD (Figure 4D). Hence, the CuMV_TT_–RBD vaccine candidate was able to induce high levels of SARS-CoV-2-neutralizing antibodies, whereas sera from mice immunized with mixed CuMV_TT_ and RBD did not show any neutralizing activity.

## 4. Discussion

The immune response to the SARS-CoV-2 infection is initiated by innate immune activation, followed by antigen-specific T and B cell responses [30]. The most important mechanism protecting against reinfection is the presence of virus-neutralizing antibodies, which is similar for almost all viruses causing acute disease rapidly causing pathogen clearance [29]. Here, we demonstrate that the display of the RBD on CuMV_TT_ VLPs induced high levels of IgG antibodies that bind recombinant RBD and the spike protein. These antibodies were functionally validated as they can block the interaction of the RBD with ACE2, the receptor for SARS-CoV-2. Such competitive inhibition is an important feature for antibody-mediated viral neutralization [23,24,31]. In accordance with the ability to block virus–receptor interaction, the antibodies induced were able to potently neutralize lentiviruses pseudotyped with the spike of SARS-CoV-2 and the actual SARS-CoV-2. 

Induction of an antibody, particularly neutralizing antibody responses to coronaviruses, is often weak and short-lived, particularly in patients with little or no symptoms, with antibody titers in most patients dropping to background levels within an estimated year [32]. Although many patients generate overall SARS-CoV-2-specific antibody responses, neutralizing antibodies often remain at low titers and appear late [33,34]. Of greatest concern may be that some patients fail to generate long-lasting antibodies [35]. There is some evidence that protection from reinfection can be very short-lived, as demonstrated by some patients who experienced COVID-19 twice, despite a proven virus-free interval in between [36,37,38]. In addition, some of the recently emerging mutated viruses of concern seem to escape antibody-mediated recognition of the RBD almost completely [39,40,41]. Thus, on average, SARS-CoV-2 triggers a curiously weak and short-lived response, while most other acute disease-causing viruses induce long-lived neutralizing antibody responses [42]. Similarly, most attenuated, replication-competent viral vaccines induce long-lived antibody responses after a single shot in the absence of disease [43].

The surface geometry of SARS-CoV-2 may offer an explanation for the unexpected short-lived antibody responses [29]. It is known that the spike protein of SARS-CoV-2 has a trimeric structure, which can interact with each other, and the receptor-binding domain (RBD), which binds to the ACE2 receptor, is associated with allosteric motion and conformational changes. However, even if this finding on the structure enhances our understanding of SARS-CoV-2 infection, it does not explain the short-lived antibody responses in some patients [44,45]. Most RNA viruses, such as orthomyxoviruses, rhabdoviruses, flaviviruses, alphaviruses, and togaviruses, to name just a few families, have highly organized and rigid surfaces, consisting of multiple copies of one or two proteins [16,46]. Such highly organized surfaces are optimal for the induction of potent and long-lived immune responses and known as a pathogen-associated structural pattern (PASP) [47]. Indeed, such viral particles have surface subunits spaced by 5–10 nm, which is optimal for inducing B cell responses. In addition to efficiently cross-linking B cell receptors [48,49], they are recognized by natural IgM, causing the activation of the classical pathway of complement. This causes the binding of viral particles to the complement receptor CD21, followed by B cell-dependent migration to and deposition on follicular dendritic cells, resulting in efficient germinal center formation [50]. Importantly, engagement of CD21 by antigen bound to complement fragments allows the induction of long-lived plasma cells, resulting in durable antibody responses [28]. In contrast, the body of the SARS-CoV-2 virion is much larger with a diameter of around 100 nm compared with 30–50 nm of most classical RNA viruses [29]. In addition, the spike protein is present at relatively low numbers, causing RBD epitopes to be spaced at much longer distances in the range of 25 nm and thus incapable of inducing an optimal antibody response. As discussed previously, while the spike forms trimers which each contains three RBDs, spaced at around 5 nm. This optimally spaced 3 RBDs per trimer may, however, not be beneficial for the immune response, as subpotimal numbers of optimally spaced epitopes inhibit rather than enhance B cell responses [29]. Hence, SARS-CoV-2 may avoid strong and long-lasting antibody responses by diluting out the neutralizing epitopes in a large amount of lipids and other membrane proteins [6,29]. The here presented strategy consisting of grafting the RBD onto highly repetitive and immunogenic virus-like particles may allow for overcoming this problem. Indeed, by displaying the RBD domain in a repetitive fashion on immunologically optimized CuMV_TT_ VLPs, one is able to shorten the distances between RBDs to the optimal spacing of 5 nm, resulting in strong antibody responses. This may allow for enhancing RBD-specific neutralizing antibody responses as desired for an efficient vaccine.

## 5. Conclusions

The here presented CuMV_TT_-based vaccine candidate relies on highly efficient expression systems and established chemical conjugation technologies and is able to induce high titers of neutralizing antibodies, which can interfere with the binding of the viral spike to ACE2 and are effective in mediating the neutralization of pseudotyped retroviruses and actual SARS-CoV-2. These results suggest that the CuMV_TT_–RBD vaccine candidate has the potential to be further developed as an effective vaccine for use in humans to protect against SARS-CoV-2 infection.

## Figures and Tables

**Figure 1 vaccines-09-00395-f001:**
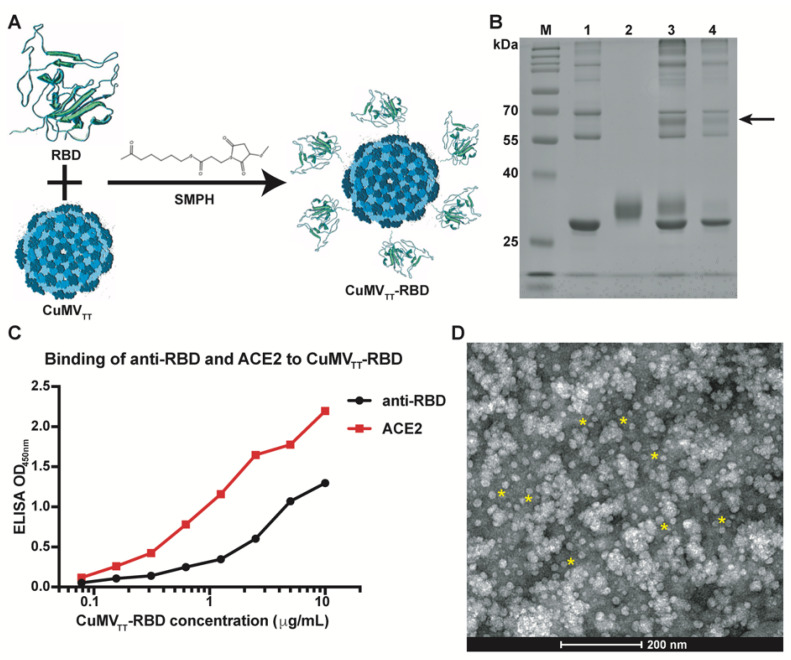
Coupling of spike–RBD with CuMV_TT_ viruslike particle (VLP). (**A**) Outline of the strategy to display RBD on CuMV_TT_ surface. (**B**) Analysis of the RBD and coupling reactions of the RBD to CuMV_TT_ by SDS-PAGE. Coupling band is indicated by an arrow. Lane 1: CuMV_TT_ linked to SMPH; lane 2: RBD; lane 3: coupled CuMV_TT_–RBD with free RBD; lane 4: coupled CuMV_TT_–RBD without free RBD. An amount of 5 μg of each sample was loaded. (**C**) Binding of the CuMV_TT_–RBD vaccine to the anti-RBD antibody and human ACE2 by ELISA. An amount of 5 μg of the anti-CuMV_TT_ antibody was coated to capture different concentrations of the CuMV_TT_–RBD vaccine. (**D**) Transmission electron microscope (TEM) image of the coupled CuMV_TT_–RBD vaccine. Yellow stars indicate the CuMV_TT_–RBD VLP.

**Figure 2 vaccines-09-00395-f002:**
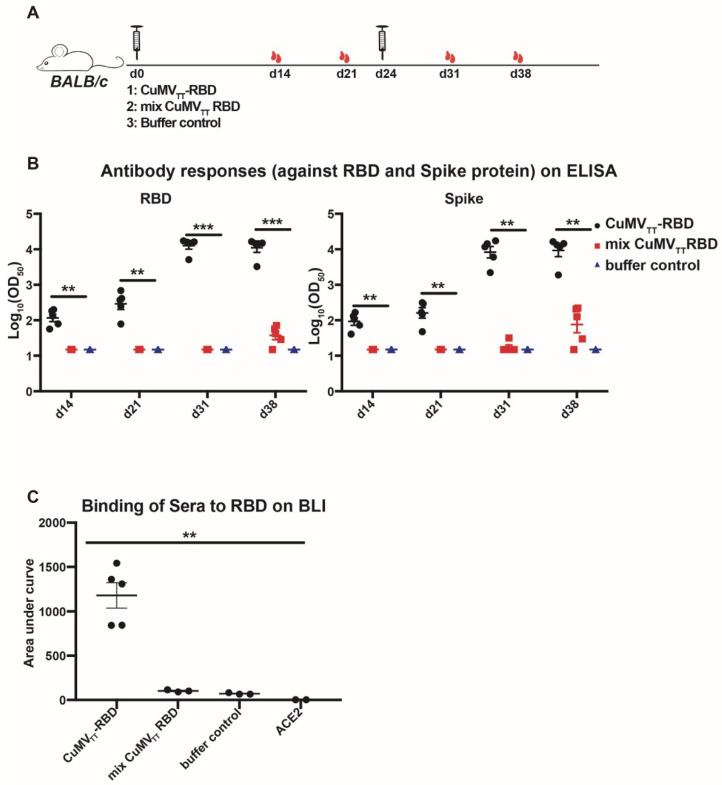
The CuMV_TT_–RBD vaccine induces high RBD-specific antibody titers. (**A**) Vaccination regimen: mice (5 per group) were vaccinated by subcutaneous injection with either 40 μg CuMV_TT_–RBD or mixed 40 μg CuMV_TT_ and 40 μg RBD or with Tris buffer (20 mM Tris-HCl, 5 mM EDTA, pH 8.0) at d0 and d24. Serum samples were harvested on days 14, 21 31, and 38 after first vaccination. (**B**) Serum samples were tested for binding to the RBD and spike protein by ELISA. An amount of 1 μg/mL of the RBD or spike protein was coated on the plates, and mouse sera were added in threefold serial dilutions starting at 1:20. The results are expressed as OD_50_ (the dilution that reached half OD_max_). Shown are a scatter plot of the IgG titers of individual mice (*n* = 5 mice). (**C**) The binding of sera to the RBD was assessed by biolayer interferometry (BLI) (octet). The sera of immunized mice at d31 after first immunization (1:20 dilution) were incubated with the RBD immobilized on a biosensor, and area under the curve (responses with time) was assessed. Shown are a scatter plot of the binding responses of individual mice (*n* > 3 mice). Statistical analysis was performed with unpaired *t*-test and *p* ≤ 0.01 (**), *p* ≤ 0.005 (***).

**Figure 3 vaccines-09-00395-f003:**
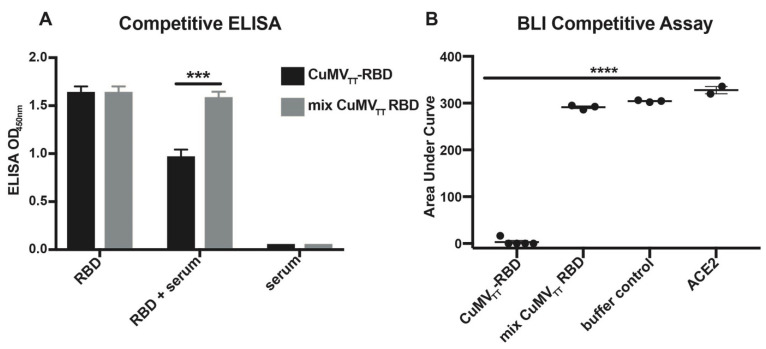
Sera of mice immunized with the CuMV_TT_–RBD vaccine inhibit the interaction of the RBD to ACE2. (**A**) Competition ELISA results using immobilized ACE2 (1 μg/mL). Five mouse sera per group (1:40 dilution, d31 after first immunization) were incubated with RBD–His (0.15 μg/mL) before adding to ACE2. (**B**) Competition BLI results using the RBD immobilized on a biosensor. Sera of immunized mice at d31 after first immunization (1:20 dilution) were used to compete for the binding of ACE2 (50 nM) to the RBD. Shown are a scatter plot of individual mice (*n* ≥ 3/group), and area under the curve (binding of ACE2 to the RBD with time) was assessed. Statistical analysis was performed with unpaired *t*-test, *p* ≤ 0.005 (***), *p* ≤ 0.001 (****).

**Figure 4 vaccines-09-00395-f004:**
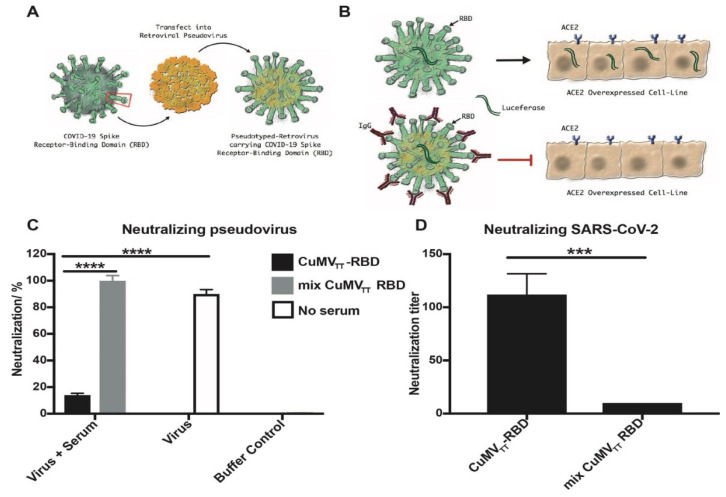
Sera of mice immunized with the CuMV_TT_–RBD vaccine neutralize pseudovirus and SARS-CoV-2. (**A**) Pseudovirus expressing the SARS-CoV-2 spike protein was generated by cotransfection of the plasmids of luciferase-expressing HIV-1 and S into HEK 293 T cells using the second-generation lentiviral packaging system. (**B**) Schematic presentation of the principle of the pseudovirus neutralization assay. (**C**) Percentage of immunized mice sera neutralizing pseudovirus. Mice sera (d31 after first vaccination) were diluted 10 times and incubated with pseudovirus. The results are expressed as percentage of the fluorescence of positive control (open bar, no serum added), which was regarded as 100%. (**D**) Neutralization titer of mice sera (d31 after first vaccination) for SARS-CoV-2. Titer is expressed as the highest dilution of the serum that fully inhibits the formation of CPE. Statistical analysis was performed with unpaired *t*-test, *p* ≤ 0.005 (***), *p* ≤ 0.001 (****).

## Data Availability

The data presented in this study are available on request from the corresponding author.

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
