# Peer review of "Development of a Vaccine against SARS-CoV-2 Based on the Receptor-Binding Domain Displayed on Virus-Like Particles"

_vaccines, 2021, doi:10.3390/vaccines9040395_

Round 1

Reviewer 1 Report

In this manuscript, Bachmann and co-workers developed a novel COVID-19 vaccine based on recombinant receptor binding domain of spike protein and cucumber mosaic virus-derived virus-like particles. They verified the effectiveness of the assay by multiple assays including ACE2 binding and generation of neutralizing antibodies in an animal model. In general, this manuscript is well written and the data is sufficient to support its conclusion. I recommend the authors clarify some details before the acceptance of this manuscript:

  1. 1B, label gels/ladders with their molecular weights.
  2. 1C, add the error bar if multiple replicates were carried out.
  3. 1D, conditions of TEM are not reported in the method section, please add.
  4. 2 - 4, clarify the details of statistical analysis, for example, what kinds of statistical tests, paired or unpaired.

Reviewer 2 Report

The manuscript “Development of A Vaccine Against SARS-Cov-2 Virus Based on The Receptor Binding Domain Displayed on Virus-Like Particles” by Lisha Zha et al. report on an important topic: a new vaccine candidate against SARS-CoV-2. To generate (fast) an effective vaccine against SARS-CoV-2 which induces production of a high level of protective antibodies is one of the most urgent issues humanity has ever dealt with. In this manuscript the authors present the development of a new vaccine using the SARS-CoV-2 Receptor Binding Domain displayed on immunologically optimized virus-like particles derived from cucumber mosaic virus. The tests of the new vaccine conducted on mice showed a promising outcome. The presented results are interesting; however, there are several issues which would require addressing before publishing (see the list below). Therefore, I recommend major revisions.  

Introduction contains outdated information about vaccine availability citing the literature from 2020 (ref 3). Currently there are 75 vaccines in clinical trials on humans (21 at the final stages of testing). Also, the Introduction contains incorrect information, such as “(…in particular due to infected individuals with very little symptoms (“spreaders”) and a long incubation time (3 weeks) combined with viral shedding long before disease onset.)”. In addition, on a number of occasions (not just in the Introduction), the referenced literature is irrelevant to the discussed subject (for example ref 2, line 43), the facts are misinterpreted or just incorrect (ref 3, line 47).

Methods are inadequately described. If the method was published before, the reference should be used. If this is a new protocol, the description would require more details. “Elisa”, “Generation of pseudovirus”, “Titration of pseudovirus”, “Pseudovirus neutralization assay” and “SARS-CoV-2 virus neutralization assay” descriptions need language editing. In addition, the TEM experiment description is missing.

The Results section is probably the shortest Results section I have ever seen (about 600 words). Paragraphs 3.1 (and 3.2 to some extent) do NOT contain any observations; just steps in the protocol are listed! For example, what is the meaning of “confirmed efficient coupling” or “perfect VLP shape”? It needs to be explained so that the reader can understand the presented data. Also, Figures are poorly described in the text and the figure legends contain very few relevant points. For example, Figure 1: panel A does NOT show design strategy; panel B does NOT have any description; panel C ELISA does NOT bind anything. Figure 2 description contains information which belongs to Methods or Results sections; panel C “vaccinated sera”? What is a meaning of the star symbol (in Fig 3 and Fig 4 as well)? The language would need some editing.

In the Discussion section one of the key points is the fixed (25nm) distance between RBD domains as a potential factor in the antibody response. Basically, there are 3 important facts ignored in this discussion: (1) proteins are dynamic structures, (2) “spikes” are actually trimeric structures and can interact with each other, and (3) RBD binding to ACE2 receptor is associated with allosteric motion and conformational change. Therefore, the argument about 25nm distance between RBDs presented on SARS-CoV-2 is entirely invalid. On the same note I disagree with the statement in line 309: “Most viruses exhibit surface epitopes in a rigid manner”.

Specific questions

  1. Why is cucumber mosaic virus an attractive platform and what advantage does it hold over for example adeno-associated virus? In my opinion it would be relevant information to add to Introduction and/or Discussion.
  2. How do conducted experiments support investigation of the RBD native conformation? The antibody binding assay does not support that claim.
  3. “Vaccine development” would suggest characterization of the entire process. I was not able to find any information on the RBD density displayed on VLP, one of the key points discussed in the Discussion section.
  4. What is a purpose of the competitive assay? How does ELISA or BLI enable the epitope recognition?
  5. Where does the information about distance between SARS-CoV-2 RBD epitopes come from? Reference 3 does not discuss this subject.

A few examples of awkward sentences, typos or words used incorrectly.

Lines 112-116: too long, unclear sentence

Line 136,137: sentences start with “and”

Lines 143, 214: confirmation?

Line 147: Illustrated?

Lines 172-173: What method was used to normalize the raw data?

Line 179: syringe filter

Line 185: monitored?

Lines 186-187: sentence needs editing

Lines 194-195: unclear sentence

Line 208: well established?

Line 255: octer?

Lines 266, 296: “real SARS-CoV-2 virus”?

Line 320: “diluting out the neutralizing 320 epitopes”?

Different abbreviations for SARS-CoV-2 used throughout the manuscript; unjustified use of capital letters throughout the manuscript

Author Response

Response to Reviewer 2 Comments

The presented results are interesting; however, there are several issues which would require addressing before publishing (see the list below). Therefore, I recommend major revisions.  

Introduction contains outdated information about vaccine availability citing the literature from 2020 (ref 3). Currently there are 75 vaccines in clinical trials on humans (21 at the final stages of testing). Also, the Introduction contains incorrect information, such as “(…in particular due to infected individuals with very little symptoms (“spreaders”) and a long incubation time (3 weeks) combined with viral shedding long before disease onset.)”. In addition, on a number of occasions (not just in the Introduction), the referenced literature is irrelevant to the discussed subject (for example ref 2, line 43), the facts are misinterpreted or just incorrect (ref 3, line 47).

Response 1: We thank the reviewer for his valuable comments. We agree with him that some references were not updated. Therefore, we have revised our references and completed with new ones. We have also updated the text according to the last findings.   

Methods are inadequately described. If the method was published before, the reference should be used. If this is a new protocol, the description would require more details. “Elisa”, “Generation of pseudovirus”, “Titration of pseudovirus”, “Pseudovirus neutralization assay” and “SARS-CoV-2 virus neutralization assay” descriptions need language editing. In addition, the TEM experiment description is missing.

Response 2: According to the reviewer’s comments the methods mentioned have been revised and completed.

The Results section is probably the shortest Results section I have ever seen (about 600 words). Paragraphs 3.1 (and 3.2 to some extent) do NOT contain any observations; just steps in the protocol are listed! For example, what is the meaning of “confirmed efficient coupling” or “perfect VLP shape”? It needs to be explained so that the reader can understand the presented data. Also, Figures are poorly described in the text and the figure legends contain very few relevant points. For example, Figure 1: panel A does NOT show design strategy; panel B does NOT have any description; panel C ELISA does NOT bind anything. Figure 2 description contains information which belongs to Methods or Results sections; panel C “vaccinated sera”? What is a meaning of the star symbol (in Fig 3 and Fig 4 as well)? The language would need some editing.

Response: We agree with the reviewer that the results as well the figure legends are quite short and would need some complement information to be more understandable. Thus, we have extended some parts in the results and completed the figure legends with more information.

In the Discussion section one of the key points is the fixed (25nm) distance between RBD domains as a potential factor in the antibody response. Basically, there are 3 important facts ignored in this discussion: (1) proteins are dynamic structures, (2) “spikes” are actually trimeric structures and can interact with each other, and (3) RBD binding to ACE2 receptor is associated with allosteric motion and conformational change. Therefore, the argument about 25nm distance between RBDs presented on SARS-CoV-2 is entirely invalid. On the same note I disagree with the statement in line 309: “Most viruses exhibit surface epitopes in a rigid manner”.

Response: Orthomyxoviruses, Rhabdoviruses, Flaviviruses, Alphaviruses, Togaviruses, to name just a few families, have all highly organized and rigid surfaces, which directly affects their interaction with the host immune system. Such surfaces are therefore considered Pathogen associated structural patterns driving B cell responses (Nat Rev Immunol. 2010 Nov;10(11):787-96. Ref. 30).  All these viruses have a specific of 5-10 nm between the surface subunits, which is optimal for inducing B cell responses while 25 is too much (Annu Rev Immunol. 1997;15:235-70,  Ref. 28). We have reviewed this in a recent paper and postulated that SARS-CoV-2 RBD induces weak and short-lived responses for this reason (NPJ Vaccines. 2021 Jan 4;6(1):2, Ref. 21). Unfortunately, we quoted the wrong review in the paper. This has been fixed now. While the reviewer’s comments (1) proteins are dynamic structures, (2) “spikes” are actually trimeric structures and can interact with each other, and (3) RBD binding to ACE2 receptor is associated with allosteric motion and conformational change are all correct, it is hard to understand why this would result in weak or short-lived responses. Hence, we think that our arguments are valuable and make give a good explanation for short-lived antibody responses. Nevertheless, we will incorporate the reviewer’s thoughts into the discussion.

Specific questions

  1. Why is cucumber mosaic virus an attractive platform and what advantage does it hold over for example adeno-associated virus? In my opinion it would be relevant information to add to Introduction and/or Discussion.

Response: Adeno-associated virus and cucumber mosaic virus are based on completely different systems. Adenovirus based vaccine such as the one developed by AstraZeneca is based on a modified version of a chimpanzee adenovirus DNA in which the gene for spike protein was inserted and can enter human cells. This vector induces not only immune responses against the vectored gene (i.e. spike protein) but also against the vector backbone itself which can compromise humoral as well as cellular mechanisms in the case of pre-existing anti-vector immunity or prime-boost vaccination. The same, if not more pronounced, is true for adeno-associated viruses, which are typically used as potential gene-therapy vectors because of low immunogenicity.

In contrast virus-like particles such as cucumber mosaic virus are non-infectious, self-assembly proteins built from several copies of the same coat protein which can be engineered chemically or genetically to present protein epitopes on their surface. The processing of VLP by antigen-presenting cells induce strong cytotoxic and humoral immune response. Even though VLPs lacks infectious genome they incorporate nucleic acids (RNA) during their assembly in the host cells (bacteria). This results in efficient activation of toll-like receptors leading to strong B cell responses, driven by B cell intrinsic TLR7/8 signaling. VLPs have several advantages including a structure similar to viruses that can induce a strong immune response, the versality of VLPs in antigen presentation and most importantly the safety as no existing immunity against VLPs such as cucumber mosaic virus is present in humans that can compromise the immune responses or can induce strong side effects as it has been recently reported for the AstraZeneca COVID vaccine.

  1. How do conducted experiments support investigation of the RBD native conformation? The antibody binding assay does not support that claim.

Response: We agree with the reviewer that there is no direct evidence for RBD to have native confirmation. However, our experiments show indirect evidence that RBD should have the right conformation: first RBD was produced in human cell line (HEK293 cell) thus allowing to produce soluble and properly folded protein as recently described in Argentinian AntiCovid Consortium, Scientific Reports 10, (2020). Second, our experiments showed that RBD on CuMVTT can bind ACE2 (Fig. 1C) and the capacity of the sera of mice immunized with CuMVTT-RBD to prevent binding of ACE2 to RBD (Fig. 3A and B). These results provide strong evidence for RBD to have the right conformation able to bind ACE2. In addition, our RBD is routinely used in ELISAs for convalescent sera, which recognized the protein very well.

  1. “Vaccine development” would suggest characterization of the entire process. I was not able to find any information on the RBD density displayed on VLP, one of the key points discussed in the Discussion section.

Response: We have recently published a review in NPJ/Vaccines (Bachmann MF, et al. NPJ/Vaccines 2021;6:2 Ref. 21) in which we explained in detail how the SARS-CoV 2 structural features may hamper a strong neutralizing-antibody response. We apologize that we forget to add this reference and have now added it in the part of the Discussion dealing with this subject. The Coupling density is now discussed in the paper.

  1. What is a purpose of the competitive assay? How does ELISA or BLI enable the epitope recognition?

Response: We performed competitive assays to determine whether anti-RBD sera from immunized mice can compete with ACE2 for binding to RBD. We developed first an ELISA where ACE2 was immobilized on the plate followed by the addition of sera of immunized mice in the presence or not of RBD. The result showed that sera of immunized strongly inhibit RBD binding to ACE2. In parallel we developed a competitive BLI assay using a different set-up (RBD is immobilized on the sensor, followed by addition of sera and then by ACE2) that confirms the ELISA assay namely that anti-RBD sera recognize an epitope that overlaps with ACE2 binding of RBD. 

  1. Where does the information about distance between SARS-CoV-2 RBD epitopes come from? Reference 3 does not discuss this subject.

Response: We completely agree with the reviewer with the fact that Ref 3 does not discuss this subject, and we apologize for this error. We have now added the reference mentioned above under point 3 which discusses extensively this subject and outlines the calculation.

A few examples of awkward sentences, typos or words used incorrectly.

Lines 112-116: too long, unclear sentence

Line 136,137: sentences start with “and”

The two and have deleted.

Lines 143, 214: confirmation?

Line 143: The sentence has been changed

Line 214: The word has changed into conformation.

Line 147: Illustrated?

Line 147: The sentence has been changed

Lines 172-173: What method was used to normalize the raw data?

Lines 172-173: The Forte Bio analysis software was used.

Line 179: syringe filter

Lines 172-173: filter syringe has been changed into syringe filter

Line 185: monitored?

Lines 172-173: The word monitored has been deleted because the whole paragraph has been edited.

Lines 186-187: sentence needs editing

This has been changed.

Lines 194-195: unclear sentence

The sentence has been changed.

Line 208: well established?

Well established has been deleted.

Line 255: octer?

Octer has been changed into octet

Lines 266, 296: “real SARS-CoV-2 virus”?

The word real has been deleted and changed into actual.

Line 320: “diluting out the neutralizing 320 epitopes”?

The end of the sentence has been modified

Reviewer 3 Report

The authors developed of A Vaccine candidate Against SARS-Cov-2 Virus Based on The Receptor Binding Domain Displayed on Virus-Like Particles, I have few concerns regarding the study per attached. Meanwhile, the discussion need extensive editing on grammar and language.

Please see attachment for more comments

Regards

Round 2

Reviewer 2 Report

I like the provided answers to my specific questions. Thank you for the effort. Most of them are informative and well structured.  I was hoping that our “offline” discussion would inspire the authors to execute the same in the Introduction and Discussion sections. Instead, the Introduction section fails to set the stage for the presented research. For example, it looks like the need for a new vaccine is required because of storage temperature and price; these two vaccine features are never discussed in this manuscript. Also, it contains irrelevant information such as ”SARS-CoV-2 shows a sequence similarity of 76% to 78% with the spike protein of SASR-CoV-1.” What is the point of this information? Why would it be a range? The Discussion section provides only a one-sided argument or ignores the existence of other scenarios. For example:

  • “There is accumulating evidence that protection from reinfection can be very short lived as demonstrated by some patients which experienced COVID-19 twice, despite a proven virus free interval in between [34].”

The conclusion based on the case study described in Reference 34 is speculative, since it is unknown if the patient was reinfected with the same strain of CoV-2. In addition, ACE2 is not the only gateway for CoV-2 into the host cells.

Mohamed S. Mohamed, Thiago C. Moulin and Helgi B. Schiöth. Sex differences in COVID-19: the role of androgens in disease severity and progression. Endocrine. 2021; 71(1): 3–8

  • “However, even if this finding on the structure enhances our understanding of SARS-CoV-2 infection, it does not explain the short-lived antibody responses in some patients [37,38].”

References 37 and 38 do not take into account that “spikes” are trimeric structures.

Some of the statements must be softened or described better in some places. The authors have agreed with me on a number of occasions in the cover letter but did not execute it in the manuscript. For example:

  1. experiments delivering “direct” and “indirect” evidence have different designs
  2. “competitive binding” is not the same as “epitope recognition”

Please, include some of the answers to my specific questions in the manuscript.  For example: advantage of CMV over AAV, which is well explained in the cover letter, but is entirely missing in the text. Also discuss (and add to a text) why mice immunized with mixed CuMVTT and RBD did not show any production of neutralizing antibodies. In addition, information about RBD density displayed on the VLP is still unavailable. Is it in the range of 5-10nm? Adding a reference is not the same as “discussed in the paper.” Even though the Results description has greatly improved it needs more work. For example, Lastly, electron microscopy analysis of the morphology of the CuMVTT-RBD vaccine demonstrated VLP shape as shown in Fig.1D.” How does it describe Fig1.D?  The particles on the image are way too small to tell anything about them. Finally, the Methods section is still missing the “Transmission electron microscope experiment” description.

My last point, please, do edit the language before next submission to avoid things like: “To limit the damage of COVID-19 “ or  “they have already delivered successful vaccines as a platform for antigen display” or “Vaccination of mice with mix CuMVTT RBD or with buffer only induced weak antibody response” or “protection from reinfection.”  Also “The disease COVID-19 is less severe than SARS and MERS, which is beneficial (...)” this is a bothersome statement. Please, change the word “beneficial” to something more appropriate.

Finally, let’s focus on the part of response which seems to create the biggest disagreement.

”While the reviewer’s comments (1) proteins are dynamic structures, (2) “spikes” are actually trimeric structures and can interact with each other, and (3) RBD binding to ACE2 receptor is associated with allosteric motion and conformational change are all correct, it is hard to understand why this would result in weak or short-lived responses. Hence, we think that our arguments are valuable and make give a good explanation for short-lived antibody responses. Nevertheless, we will incorporate the reviewer’s thoughts into the discussion.”

Before I dive into details, here is what the authors said in the Discussion at some point: “Most RNA viruses such Orthomyxoviruses, Rhabdoviruses, Flaviviruses, Alphaviruses, Togaviruses, to name just a few families, have highly organized and rigid surfaces, consisting of multiple copies of one or two proteins [15,39].” The key words here are: “rigid surface” and “multiple copies”, unlike “spikes” which are highly dynamic structures and less populated on the virus’ surface.  Here is why the authors’ argument (25nm spacing of RBD) is NOT a good explanation for short-lived antibody response.  I have prepared a figure showing localization of the receptor binding motifs in the trimeric structure of the spike (in red, green and blue). I used one of the first published structures of the trimer. Based on calculated distance between 3 RBD domains within single “spike”, I would say that RBD spacing is actually spot on, fitting in the 5-10nm range, optimal for inducing B cells response. This is just based on a “static” view (cryo-EM structure). Now, if we take into the account interactions between “spikes” we can probably find more areas temporarily (or permanently) “enriched” in epitopes. This is specifically important since there are many epitopes in different regions of the spike protein. There are several epitopes (linear and conformational) present in RBD alone. In fact, the dominant RBD epitopes are actually of the conformational type. For more information on “spike” dynamics see the following publications:

  1. Turoňová B, Sikora M, Schürmann C, Hagen WJH, Welsch S, Blanc FEC, Sören von Bülow S, Gecht M, Bagola K, Hörner C, v Zandbergen G, Mosalaganti S, Schwarz A, Covino R, Mühlebach MD, Hummer G, Krijnse Locker J, Beck M.In situ structural analysis of SARS-CoV-2 spike reveals flexibility mediated by three hinges. Science; 18.08.2020
  2. Ghorbani, M. et al. (2020). Exploring Dynamics and Network Analysis Of Spike Glycoprotein Of SARS-COV-2. bioRxiv preprint. doi: https://doi.org/10.1101/2020.09.28.317206. https://www.biorxiv.org/content/10.1101/2020.09.28.317206v1
  3. Structure, Dynamics, Receptor Binding, and Antibody Binding of Fully-glycosylated Full-length SARS-CoV-2 Spike Protein in a Viral Membrane Yeol Kyo Choi, Yiwei Cao, Martin Frank, Hyeonuk Woo, Sang-Jun Park, Min Sun Yeom, Tristan I. Croll, Chaok Seok, Wonpil Im bioRxiv 2020.10.18.343715; doi: https://doi.org/10.1101/2020.10.18.343715,  https://www.biorxiv.org/content/10.1101/2020.10.18.343715v1

Figure. Distance between receptor binding motifs in the trimeric S-protein structure of the SARS-CoV-2. Binding motifs highlighted in red, green and blue. PDB ID: 6VYB.

C. Walls, Y.-J. Park, M. A. Tortorici, A. Wall, A. T. McGuire, D. Veesler, Structure, Function, and Antigenicity of the SARS-CoV-2 Spike Glycoprotein. Cell 181, 281–292.e6 (2020).

Author Response

Response to Reviewer 2 Comment

I like the provided answers to my specific questions. Thank you for the effort. Most of them are informative and well structured.  I was hoping that our “offline” discussion would inspire the authors to execute the same in the Introduction and Discussion sections. Instead, the Introduction section fails to set the stage for the presented research. For example, it looks like the need for a new vaccine is required because of storage temperature and price; these two vaccine features are never discussed in this manuscript.

Response 1: We thank the reviewer for his positive feedback regarding our corrections. To meet the reviewer’s comments, we added the following text to the introduction:

However, despite the progress of these vaccines, there are many obstacles hindering their rapid and efficient use specially in developing countries such as storage requirements at low temperatures -80°C (Pfizer/BioNTech vaccine) as well as the high price of some of them. Hence, next generation vaccines should be more affordable and handling should be easier compared to current RNA-based vaccines. In addition, as SARS-CoV-2 may stay with us for many years or decades, it would be beneficial if the vaccine could be applied multiple times to the same individual which is a major problem for adenoviruses. Additional important requirement for effective vaccines is their efficacy in old people, all ethnicities and despite comorbidities (Mohamed et al. Endocrine 2021, Ref. 14 in the manuscript). Because of these variables, not all vaccines may be ideal in all situations and there is room for different vaccine types optimized for specific settings.

Also, it contains irrelevant information such as ”SARS-CoV-2 shows a sequence similarity of 76% to 78% with the spike protein of SASR-CoV-1.” What is the point of this information? Why would it be a range?

How closely related the two viruses are is of considerable interest, since a) they are quite different from their epidemiology but b) use the same receptor. It is a range as viruses are typically a quasi-species with multiple sequences in given viral pool. In addition, the point concerning the sequence similarity was a requisite of Reviewer 3. As he wanted to know exactly how much the homology percentage was, we have added this information.

The Discussion section provides only a one-sided argument or ignores the existence of other scenarios. For example:

  • “There is accumulating evidence that protection from reinfection can be very short lived as demonstrated by some patients which experienced COVID-19 twice, despite a proven virus free interval in between [34].”

The conclusion based on the case study described in Reference 34 is speculative, since it is unknown if the patient was reinfected with the same strain of CoV-2. In addition, ACE2 is not the only gateway for CoV-2 into the host cells.

We agree with the reviewer that Ref. 34 might not be the best reference for showing that reinfection might happen because the study is limited to one patient. However, there are other examples showing that protective immunity is short lived (Edridge A et al. Nature medicine. 2020; 26:1691-1693; https://www.globaldata.com/antibodies-sars-cov-2-wane-time-suggesting-immunity-may-short-lived-says-globaldata/. We have added additional references to support our statement.

In addition, we added that the recently described mutated viruses seem to escape immunity as convalescent sera fail to neutralize the virus (Vogel, M et al. BioRxiv. 2021 DOI: 10.1101/2021.03.04.433887).

We agree with the review that ACE2 is not the only gateway for CoV-2 into the host cells and that SARS-CoV-2 may exploit additional receptors for infection such as receptors of the innate immune system, including C-lectin type receptors (CLR), toll-like receptors (TLR) and neuropilin-1 (NRP1), as well as the non-immune receptor glucose regulated protein 78 (GRP78). Interestingly, it has even been shown that carbohydrate moieties clustered on the surface of the S protein may also drive receptor-dependent internalization, thereby accentuating severe immunopathological inflammation, and allow for systemic spread of infection, independent of ACE2. However, in our study we focused on the interaction between the spike protein and its receptor ACE2 as many studies have shown a high correlation between neutralizing antibodies against the spike and ACE2. For example, a recent longitudinal study from a cohort of 140 samples SARS-CoV-2 confirmed patients, has shown a high correlation between neutralizing antibodies and COVID-19 severity. (Legros, V et al. Cellular and Molecular Immunology. 2021; 18:318-321.

For completeness, ae added the following reference: Mohamed S. Mohamed, Thiago C. Moulin and Helgi B. Schiöth. Sex differences in COVID-19: the role of androgens in disease severity and progression. Endocrine. 2021; 71(1): 3–8

For completeness, we added this reference in the introduction (page 3 Ref. 14)

  • “However, even if this finding on the structure enhances our understanding of SARS-CoV-2 infection, it does not explain the short-lived antibody responses in some patients [37,38].”

References 37 and 38 do not take into account that “spikes” are trimeric structures.

Response 3: The reference 37 deal with the conformation structure of the spike protein and the role of the fusion peptide in subunit S2 in the stabilization of the trimer. The reference 38 also looks at the structure by cryo-electron microscopy full-length S protein, representing its prefusion (2.9Å resolution) and postfusion (3.0Å resolution) conformations, respectively. It is true that both references do not directly describe the trimeric structure of the spike protein therefore we have added two new references namely Sternberg A et al., Life Sciences. 2020; 257 and Ke Z, Oton J et al. Nature. 2020 2020 Dec;588(7838):498-502. doi: 10.1038/s41586-020-2665-2 which both showed structural features of the spike protein using cryo-electron microscopy. 

Some of the statements must be softened or described better in some places. The authors have agreed with me on a number of occasions in the cover letter but did not execute it in the manuscript. For example: 

  1. experiments delivering “direct” and “indirect” evidence have different designs
  2. “competitive binding” is not the same as “epitope recognition”

Response 4: We agree with the reviewer that experiments deliver direct and indirect evidence have different designs. Thus, in order to make it clearer we have now subdivided the methods part in different sections describing each method in details.

We agree that competitive binding is not per se the same as epitope recognition but this is not what we meant. In our approach we used two different competitive assays one based on ELISA and another one based on BLI, with both assays we could show that patient antibodies can compete with ACE2 to bind RBD and thereby might overlap with or is in close vicinity of the ACE2 binding site of SARS-CoV-2 RBD. To this end, we have rephrased our sentence to make our statement clearer (see Results page 10 section 3.3, last sentence).

 Please, include some of the answers to my specific questions in the manuscript.  For example: advantage of CMV over AAV, which is well explained in the cover letter, but is entirely missing in the text. Also discuss (and add to a text) why mice immunized with mixed CuMVTT and RBD did not show any production of neutralizing antibodies. In addition, information about RBD density displayed on the VLP is still unavailable. Is it in the range of 5-10nm? Adding a reference is not the same as “discussed in the paper.” Even though the Results description has greatly improved it needs more work. For example, Lastly, electron microscopy analysis of the morphology of the CuMVTT-RBD vaccine demonstrated VLP shape as shown in Fig.1D.” How does it describe Fig1.D?  The particles on the image are way too small to tell anything about them. Finally, the Methods section is still missing the “Transmission electron microscope experiment” description.

Response 5: According to reviewer’s request we have incorporated the answers to some of his specific questions (please see introduction page 4 end of the first paragraph). We have also added the RBD density (please see introduction page 3 last paragraph). We agree that the resolution of Fig. 1D was not optimal and the result description might be extended therefore we modified Fig. 1 D on which the VLPs are indicated with yellow stars. We also modified the text in the result description accordingly. In response to the reviewer's comment of his last revision, we added the description of the TEM method in the second part of section 2.3. However, for the sake of clarity we have transferred the TEM method in a new section (2.4 Electron microscopy), hoping that this time is clearer.

My last point, please, do edit the language before next submission to avoid things like: “To limit the damage of COVID-19 “ or  “they have already delivered successful vaccines as a platform for antigen display” or “Vaccination of mice with mix CuMVTT RBD or with buffer only induced weak antibody response” or “protection from reinfection.”  Also “The disease COVID-19 is less severe than SARS and MERS, which is beneficial (...)” this is a bothersome statement. Please, change the word “beneficial” to something more appropriate.

Response 6: we have changed the language according to the reviewer's request:

  • “To limit the damage of COVID-19” was changed to: “To prevent further spread of COVID-19”.
  • “they have already delivered successful vaccines as a platform for antigen display“ was changed to “they have already delivered successful vaccine delivery platforms”.
  • “Vaccination of mice with mix CuMVTT RBD or with buffer only induced weak antibody response“was changed to “Vaccination of mice with a mixture of CuMVTT and RBD or with buffer only induced weak antibody response.
  • “Beneficial” was changed to “advantageous from an epidemiological point of view“.

Finally, let’s focus on the part of response which seems to create the biggest disagreement.

”While the reviewer’s comments (1) proteins are dynamic structures, (2) “spikes” are actually trimeric structures and can interact with each other, and (3) RBD binding to ACE2 receptor is associated with allosteric motion and conformational change are all correct, it is hard to understand why this would result in weak or short-lived responses. Hence, we think that our arguments are valuable and make give a good explanation for short-lived antibody responses. Nevertheless, we will incorporate the reviewer’s thoughts into the discussion.” 

Before I dive into details, here is what the authors said in the Discussion at some point: “Most RNA viruses such Orthomyxoviruses, Rhabdoviruses, Flaviviruses, Alphaviruses, Togaviruses, to name just a few families, have highly organized and rigid surfaces, consisting of multiple copies of one or two proteins [15,39].” The key words here are: “rigid surface” and “multiple copies”, unlike “spikes” which are highly dynamic structures and less populated on the virus’ surface.  Here is why the authors’ argument (25nm spacing of RBD) is NOT a good explanation for short-lived antibody response.  I have prepared a figure showing localization of the receptor binding motifs in the trimeric structure of the spike (in red, green and blue). I used one of the first published structures of the trimer. Based on calculated distance between 3 RBD domains within single “spike”, I would say that RBD spacing is actually spot on, fitting in the 5-10nm range, optimal for inducing B cells response. This is just based on a “static” view (cryo-EM structure). Now, if we take into the account interactions between “spikes” we can probably find more areas temporarily (or permanently) “enriched” in epitopes. This is specifically important since there are many epitopes in different regions of the spike protein. There are several epitopes (linear and conformational) present in RBD alone. In fact, the dominant RBD epitopes are actually of the conformational type. For more information on “spike” dynamics see the following publications:

  1. Turoňová B, Sikora M, Schürmann C, Hagen WJH, Welsch S, Blanc FEC, Sören von Bülow S, Gecht M, Bagola K, Hörner C, v Zandbergen G, Mosalaganti S, Schwarz A, Covino R, Mühlebach MD, Hummer G, Krijnse Locker J, Beck M.In situ structural analysis of SARS-CoV-2 spike reveals flexibility mediated by three hinges. Science; 18.08.2020
  2. Ghorbani, M. et al. (2020). Exploring Dynamics and Network Analysis Of Spike Glycoprotein Of SARS-COV-2. bioRxiv preprint. doi: https://doi.org/10.1101/2020.09.28.317206. https://www.biorxiv.org/content/10.1101/2020.09.28.317206v1
  3. Structure, Dynamics, Receptor Binding, and Antibody Binding of Fully-glycosylated Full-length SARS-CoV-2 Spike Protein in a Viral Membrane Yeol Kyo Choi, Yiwei Cao, Martin Frank, Hyeonuk Woo, Sang-Jun Park, Min Sun Yeom, Tristan I. Croll, Chaok Seok, Wonpil Im bioRxiv 2020.10.18.343715; doi: https://doi.org/10.1101/2020.10.18.343715,  https://www.biorxiv.org/content/10.1101/2020.10.18.343715v1

Figure. Distance between receptor binding motifs in the trimeric S-protein structure of the SARS-CoV-2. Binding motifs highlighted in red, green and blue. PDB ID: 6VYB.

  1. Walls, Y.-J. Park, M. A. Tortorici, A. Wall, A. T. McGuire, D. Veesler, Structure, Function, and Antigenicity of the SARS-CoV-2 Spike Glycoprotein. Cell 181, 281–292.e6 (2020).

We appreciate the desire of the reviewer to disprove our point of epitope spacing; it is certainly a lively discussion. Nevertheless, the reviewer should appreciate that exactly this hypothesis was published in a peer-reviewed Journal of high quality in the field of vaccines (npj Vaccines (2021)6:2; https://doi.org/10.1038/s41541-020-00264-6 Ref. 29 in the manuscript). In this article, we took into account that spike is a trimer. And came to the following conclusion (quoted from the mentioned paper):

«The S protein forms a trimer. Consequently, the RBD will display three identical epitopes favorably spaced by about 3–5 nm. As discussed above, three epitopes are, however, not enough to optimally activate B cells. On the contrary, epitopes occurring in low numbers inhibit, rather than activate, B-cell responses. Indeed, Dintzis et al. (J. Immunol. 131, 2196–2203 (1983) concluded that increasing epitope density in a molecular structure increases its immunogenicity if the threshold  number of ∼20 is reached. In contrast, increasing the density in a molecular structure below the threshold number increases its tolerogenicity . Thus, trimeric RBD may reduce rather than increase neutralizing antibody responses.»

Hence, we argue that our structural argument has some merits and we would prefer to leave it in.

Reviewer 3 Report

Dear authors,

Thanks to consider my comments. Good luck.

Author Response

We thank the reviewer for his positive feedback.

Best regards

Martin Bachmann

Round 3

Reviewer 2 Report

I would like to thank the authors for such a detailed and to the point response to my comments. This effort is definitely reflected in the manuscript quality. The Introduction section adequately sets the stage for the presented research and clearly defines the goals to be attained. The Results and Discussion have greatly improved as well. Even though one side of the argument is still underappreciated (protein dynamics) I like the provided explanation. This is a truly fascinating subject to study; I am looking forward to reading about the next generation of experiments. My recommendation is to accept the manuscript in its present form.